# The Molecular Interplay between Human Oncoviruses and Telomerase in Cancer Development

**DOI:** 10.3390/cancers14215257

**Published:** 2022-10-26

**Authors:** Maria Lina Tornesello, Andrea Cerasuolo, Noemy Starita, Anna Lucia Tornesello, Patrizia Bonelli, Franca Maria Tuccillo, Luigi Buonaguro, Maria G. Isaguliants, Franco M. Buonaguro

**Affiliations:** 1Molecular Biology and Viral Oncology Unit, Istituto Nazionale Tumori IRCCS “Fondazione G. Pascale”, Via Mariano Semmola, 80131 Napoli, Italy; 2Cancer Immunoregulation Unit, Istituto Nazionale Tumori IRCCS “Fondazione G. Pascale”, Via Mariano Semmola, 80131 Napoli, Italy; 3Research Department, Riga Stradins University, LV-1007 Riga, Latvia

**Keywords:** telomerase reverse transcriptase, TERT, TERT promoter, TERTp, human papillomavirus, HPV, Epstein–Barr virus, EBV, Kaposi’s sarcoma-associated herpesvirus, HHV-8, hepatitis B virus, HBV, hepatitis C virus, HCV, human T-cell leukemia virus-1, HTLV-1

## Abstract

**Simple Summary:**

The expression of the telomerase reverse transcriptase (TERT) gene is commonly repressed in terminally differentiated somatic cells and becomes reactivated in the large majority of tumors. Oncogenic viruses have evolved multiple strategies to subvert the telomerase function in the host cells. Viruses may promote TERT transcription through the binding of their oncoproteins to cis-regulatory elements in the gene promoter or by integrating their genomes nearby TERT locus. Other viruses cause telomerase activation via epigenetic mechanisms that contribute to the maintenance of long telomeres and cellular immortality. This review will outline recent findings on the strategies employed by viruses to deregulate telomerase activity and telomere length and to promote cancer development.

**Abstract:**

Human oncoviruses are able to subvert telomerase function in cancer cells through multiple strategies. The activity of the catalytic subunit of telomerase (TERT) is universally enhanced in virus-related cancers. Viral oncoproteins, such as high-risk human papillomavirus (HPV) E6, Epstein–Barr virus (EBV) LMP1, Kaposi’s sarcoma-associated herpesvirus (HHV-8) LANA, hepatitis B virus (HBV) HBVx, hepatitis C virus (HCV) core protein and human T-cell leukemia virus-1 (HTLV-1) Tax protein, interact with regulatory elements in the infected cells and contribute to the transcriptional activation of TERT gene. Specifically, viral oncoproteins have been shown to bind TERT promoter, to induce post-transcriptional alterations of TERT mRNA and to cause epigenetic modifications, which have important effects on the regulation of telomeric and extra-telomeric functions of the telomerase. Other viruses, such as herpesviruses, operate by integrating their genomes within the telomeres or by inducing alternative lengthening of telomeres (ALT) in non-ALT cells. In this review, we recapitulate on recent findings on virus–telomerase/telomeres interplay and the importance of TERT-related oncogenic pathways activated by cancer-causing viruses.

## 1. Introduction

More than 12% of human cancers are caused by seven oncogenic viruses, including high-risk human papillomaviruses (HPV), Epstein–Barr virus (EBV), hepatitis B (HBV) and C (HCV) viruses, human T-cell lymphotropic virus 1 (HTLV-1), Kaposi’s sarcoma herpesvirus (HHV8), and Merkel cell polyomavirus (MCPyV) [1]. Although human oncoviruses belong to diverse virus families and hold specific tissue tropism as well as distinct replication mechanisms, they have similar pathogenic activities [2,3]. All oncoviruses promote cell survival and transformation through several common features that include the ability to establish long-lasting persistent infections, to cause chronic insults and deregulation of metabolic pathways, and to trigger the progressive accumulation of genetic damages and immune escape [3].

The telomerase complex, by extending the ends of chromosomes during each cell division, plays a key role in the evasion of cellular senescence and promotion of replicative immortality of cancer cells [4]. The telomerase activity is restored in more than 75% of human tumors via multiple mechanisms including the aberrant expression of transcription factors binding to the TERT promoter, TERT gene amplification, post-transcription modifications of TERT mRNA, as well as activating mutations and methylation of TERT regulatory regions [5,6].

Tumor viruses have the ability to enhance telomerase activity and telomere length and to contribute to the unlimited proliferation and transformation of chronically infected cells [7]. Of note, several oncoproteins encoded by human tumor viruses, such as HPV E6, EBV LMP1 and HHV-8 LANA, transactivate TERT gene leading to the immortalization and transformation of human epithelial cells [8,9,10]. The genomes of some viruses, such as HBV, have been frequently found integrated nearby the TERT gene locus causing enhanced TERT expression [11]. Furthermore, the insertion of herpesviruses DNA within telomeres of latently infected cells, facilitated by the high homology between the virus and human telomere sequences, is crucial for tumor formation and reactivation of latent infections [12,13].

This review summarizes the diverse mechanisms operated by six oncoviruses to enhance telomerase activity in the human cells and the possible oncogenic mechanisms involved in the promotion of cancer.

## 2. TERT Expression and Telomerase Activities

Telomeric DNA repeats (TTAGGG), located at the ends of chromosomes, are essential to prevent end-to-end chromosomal fusions and to maintain genome integrity during successive cycles of cell division [4,14]. The telomerase complex, which elongates and protects telomeres, is composed of the telomerase reverse-transcriptase holoenzyme (TERT), the RNA template component (TERC) and the dyskerin (DKC1)-NOP10-NHP2-GAR1 tetrameric complex [15,16]. Moreover, the telomere ends are protected from DNA damage response by a shelterin complex formed by six proteins, namely telomeric repeat binding protein 1 (TRF1), telomeric repeat binding protein 2 (TRF2), TERF1 interacting nuclear factor 2, member of RAS oncogene family RAP1, tripeptidyl peptidase 1 (TPP1) and protection of telomeres 1 (POT1) [15]. TERC and other telomerase components are ubiquitously and constitutively expressed in human cells, while TERT expression, which is the rate-limiting catalytic subunit of telomerase, is restricted to embryos and stem cells [17]. The TERT gene becomes repressed in the majority of somatic cells leading to a progressive shortening of telomerases during successive cell divisions and replicative senescence (the Hayflick limit) [18].

The tight regulation of TERT levels is maintained by the complex cooperation of numerous transcription factors acting as repressors or activators and by signaling pathways. However, in cancer cells the stringent control of TERT transcription is disrupted by the aberrant expression of TERT promoter activators, primarily MYC, the first oncoprotein demonstrated to induce the telomerase expression, and by nuclear factor kappa B (NF-κB) signaling that is considered the master regulator of TERT activation in cancer cells [19,20,21]. Other modalities of telomerase re-activation in tumors include chromosomal rearrangements, TERT gene amplification and TERT promoter hypermethylation [22,23]. Short telomeres also induce TERT expression by determining the detachment of the 5p sub-telomeric region from the TERT locus, which is a TERT transcription repression mechanism operated only by long telomeres [24].

More recently, single nucleotide mutations in the proximal promoter of the TERT gene, particularly at positions-124A and -146A upstream of the ATG start site, have been shown to generate de novo consensus binding motifs (GGAA) for the E-Twenty-Six family transcription factors causing the aberrant expression of telomerase [23,25,26,27]. These mutations are highly frequent in a vast majority of cancers including glioblastoma (80–90%), melanoma (70%), basal cell carcinoma (70%), hepatocellular carcinoma (60%), bladder cancers (60%), penile carcinoma (53%), conjunctival carcinoma (46%) and oral cancer (above 30%) [28,29,30,31,32,33].

In addition to the telomere-lengthening function, TERT can act as a transcription factor contributing to the regulation of multiple pathways involved in the physiological processes, such as cell renewal and tissue homeostasis; additionally it contributes to pathologic conditions, such as tumor formation and resistance to treatments [34]. TERT has been shown to bind NF-κB p65 subunit and to regulate NF-κB-dependent genes, including matrix metalloproteinase (MMP) genes as well as to activate Wnt/β-catenin signaling in gastric and prostate cancers [5]. In particular, TERT has been shown to bind SWI/SNF related, matrix associated, actin dependent regulator of chromatin, subfamily A, member 4 (SMARCA4) factor, which is a chromatin remodeler, and to induce the expression of Wnt-responsive genes such as MYC proto-oncogene (MYC) and the vascular endothelial growth factor (VEGF) thus promoting cell transformation [35,36,37]. On the other hand, the Wnt/β-catenin signaling has been reported to regulate the telomerase activity and the acquisition of cancer stem cell-like phenotype in the radioresistant nasopharyngeal carcinoma cell line CNE-2R through a positive feedback loop [38]. Other non-canonical functions of TERT include the ability to induce overexpression of the epidermal growth factor receptor (EGFR) in human mammary epithelial cells and DNA methyltransferases (DNMTs) in human fibroblasts as well as the downregulation of pro-apoptotic genes in diverse human cancer-derived cell lines [39,40,41,42]. Specifically, TERT-induced VEGF expression promotes angiogenesis, while TERT-related DNMTs’ expression determines the abnormal methylation and silencing of Phosphatase and tensin homolog (PTEN) promoter [42,43]. PTEN is a tumor suppressor that inhibits phosphatidylinositol-4,5-biphosphate 3-kinase (PI3K)/AKT-serine/threonine kinase (AKT) signaling; consequently, TERT-induced PTEN silencing causes increased AKT activity and cell survival and proliferation.

Furthermore, the telomerase activity has been shown to play a crucial role in the gradual development of T-cell lineage. Indeed, TERT level in thymocytes is nearly 30 times higher than that found in peripheral blood resting-T cells, suggesting that T-cell maturation is associated with TERT downregulation, similar to other differentiated somatic cells [44]. Activation of T cells by the engagement of T-antigen receptors is linked to an increase in TERT mRNA and telomerase activity [45]. However, individuals with chronic viral infections, such as EBV and HCV infections, have a greater proportion of exhausted virus-specific T cells, characterized by short telomeres and reduced telomerase activity [46].

In conclusion, the mechanisms underlying telomeric and non-telomeric functions of telomerase in cell transformation and immune regulation are controlled by complex mechanisms and have yet to be fully exploited.

### 2.1. Human Papillomavirus and Telomerase Activity in HPV-Related Tumours

The Papillomaviridae family comprises a large group of small, non-enveloped viruses with approximately 8-kilobase double-stranded circular DNA genomes that are classified in five genera (α, β, γ, μ, and ν) [47]. The α genus includes 13 viral genotypes (HPV 16, 18, 31, 33, 35, 39, 45, 51, 52, 56, 58, 66, and 68), defined as high risk HPVs and classified as carcinogenic to humans (group 1) by the International Agency on Cancer Research (IARC) [48]. The high-risk HPVs are the causative agents of nearly all cases of cervical carcinoma and a significant fraction of cancers arising in the mucosal squamous epithelia of the oropharynx and the lower genital tract [49]. Overall, HPV causes over 530,000 cases of cervical cancer and 113,000 cases of oropharyngeal, vulvar, vaginal, penile and anal cancers annually in the world, accounting for 29.5% of all infection-related tumors [50].

The early proteins E6 and E7 encoded by high risk HPVs are the most important factors in virus carcinogenesis for their ability to abrogate the function of tumor suppressors and to interact with multiple cell factors involved in the regulation of cell proliferation and apoptosis [51,52,53,54,55]. In particular, the HPV E6 forms a complex with E3 Ubiquitin Ligase E6-Associated Protein (E6AP) that targets p53 for degradation, while HPV E7 activates and recruits calpain-1, which cleaves the C-terminal domain (Rb-Lys^810^) of RB-transcriptional corepressor (RB) thus facilitating ubiquitination mediated by cullin E3 ligases [56,57,58,59].

Telomerase was found expressed in all HPV-related tumors and the TERT level is proportional to the severity of the neoplastic grade in cervical lesions [60]. As shown in Figure 1, the HPV E6 oncoproteins are able to promote telomerase activity by multiple mechanisms involving the regulation of transcriptional, epigenetic and post-transcriptional processes [61,62,63]. Early studies showed that the expression of HPV16 E6 in early-passage human keratinocytes induced high levels of telomerase and that such effect was independent from the HPV E6/E6AP degradation of p53 [9,64]. Moreover, TERT levels and telomerase activity were significantly higher in human keratinocytes expressing both HPV E6 and E7 oncogenes compared to cells expressing HPV E6 alone [65]. The minimal E6-responsive region in the TERT promoter was found located within the 258-bp sequence, proximal to the ATG start site [65]. This region, defined as the TERT core promoter, contains several E-boxes/X-boxes and GC-boxes, which are cis-regulatory sequences preferentially bound by the transcription factors MYC/MAX/MAD1 and SP1, respectively [66,67]. Several studies have demonstrated that MYC can activate TERT expression through the formation of MYC/MAX heterodimers, which displace MAD1/MAX repressors on the E-boxes [21,68,69]. The E6/E6AP complex has been shown to cooperate with MYC in the stabilization of MYC/MAX heterodimers, which causes the shift of MAD1/MAX repressor and enhanced TERT transcription and telomerase activity [65,70,71,72,73]. The mutation of MYC-binding sites did not result in the complete abrogation of TERT expression, which was instead obtained by the disruption of both MYC- and SP1-binding sites. These results suggested a complex mechanism of E6-dependent TERT promoter activation involving a cooperative action of transcription factors binding simultaneously to the MYC and SP1 cis-elements [65,67].

Further studies demonstrated that HPV E6 in concert with E6AP binds nuclear transcription factor, X-Box binding 1, isoform 91 (NFX1-91), a transcriptional repressor of the TERT promoter, causing its ubiquitination and degradation [74,75,76,77]. The NFX1-91 interacts with the mSin3A-histone deacetylase complex and its removal causes the overexpression of histone acetyltransferase, the increase in histone protein acetylation and enhancement of TERT transcription [78]. The HPV16 E6 has also been demonstrated to cooperate with a splice variant of NFX1-91, namely NFX1-123, as well as with cellular RNA-processing proteins, such as cytoplasmic poly(A) binding proteins (PABPCs), and to cause the increase in telomerase activity through the post-transcriptional stabilization of TERT mRNA in human keratinocytes [79,80]. The study of NFX1-123 transcripts in biological samples showed that it is highly expressed in cervical pre-cancer lesions, invasive carcinoma and derived cell lines [81]. On the other end, transient knockdown of NFX1-123 in HPV16-positive SiHa cells led to pronounced TERT reduction and slowing of cell growth [81].

In addition, HPV16 E6/E6AP complex has been reported to cause the acetylation of histone H3 at TERT promoter causing elevated TERT mRNA levels and telomerase activity particularly in early passage keratinocytes expressing HPV E6 [77]. Long term culture of E6-transduced keratinocytes caused histone acetylation at the TERT promoter and decrease in the coactivator/acetyl transferase p300 which is targeted by E6 [77].

More recently, HPV E6 has been reported to promote TERT expression through the destabilization of the tumor suppressor TIP60, a histone acetyltransferase enzyme that regulates gene transcription [82]. TIP60 is involved in the SP1 acetylation at the residue K639, which inhibits SP1 binding to the TERT promoter and determines repression of TERT gene transcription [82]. TIP60 has also a role in the repression of HPV E6 expression through its binding to the HPV early promoter, the histones’ acetylation and recruitment of bromodomain containing 4 (BRD4) factor [83]. However, the E6 proteins encoded by low- and high-risk HPV genotypes interact with E3 ubiquitin ligase EDD1 and cause the polyubiquitination and proteasomal degradation of TIP60, thus determining transcriptional activation of both HPV E6 and the TERT gene [84].

Integration of HPV genomes into the human DNA is considered a crucial event in the HPV-related carcinogenesis [85,86]. The process involves linearization of HPV DNA episomes, disruption of E2 genes and concomitant loss of E2 negative regulatory function triggering increased levels of E6 and E7 oncoproteins [87,88]. The HPV integration is frequently associated with chromosomal aberrations, extensive genomic instability and copy number variations [89]. In some cervical cancer biopsies and derived cell lines, the HPV DNA has been shown to integrate near the TERT locus with viral enhancers activating TERT promoter and increased telomerase expression [90].

Furthermore, HPV E6 protein has been shown to induce telomerase activation via a post-translational mechanism based on the direct interaction of E6 protein with TERT protein and association with telomerase complexes and telomeric DNA [61].

Of note, TERT expression and telomerase activity in cervical samples of women diagnosed with ASCUS (abnormal squamous cells of undetermined significance), LSIL (low-grade squamous intraepithelial lesion), HSIL (high-grade squamous intraepithelial lesion) and invasive cervical carcinoma have been shown to increase with disease severity [60,91,92]. Similar to cervical neoplasia, head and neck cancers related to HPV infection show higher levels of TERT expression compared to those negative for HPV [93,94,95]. On the basis of such results, telomerase may be considered a useful biomarker for the early detection of cervical lesions as well as an ideal target for anti-cancer therapies.

### 2.2. Epstein-Barr Virus and Telomerase Activity in EBV-Related Tumours

The EBV or human gamma herpesvirus 4 is a ubiquitous virus of the Herpesviridae family, containing a genome of 184 kb linear double-stranded DNA, which can infect both epithelial cells and B cells [96]. Following primary infection, EBV performs a short lytic program and then primarily establishes a persistent lifelong infection in almost all human subjects. A few latently infected cells are able to switch from the latent stage into the lytic cycle to produce infectious virus particles [97].

EBV infection can cause Burkitt’s lymphoma and nasopharyngeal carcinoma, as well as, to a lesser extent, other human malignancies, such as Hodgkin disease, gastric cancer, diffuse large B-cell lymphoma and extranodal NK/T-cell lymphoma [98,99]. The global burden of EBV-related tumors from these six cancers accounted for between 239,700 to 357,900 new cases and 137,900 to 208,700 deaths in 2020 [100].

EBV-related tumorigenesis is associated with the establishment of latency programs, such as EBV latency I, II or III, and the differential expression of latent viral proteins in virus-infected epithelial tumors and lymphoblastoid cell lines [97]. In particular, the EBV latency I with selective expression of EBV nuclear antigen (EBNA)-1 is frequently identified in Burkitt’s lymphoma cells. The EBV latency program II, characterized by the expression of EBNA-1 and latent membrane proteins LMP-1, LMP-2A and LMP-2B, is typical of Hodgkin lymphoma and nasopharyngeal carcinoma cells. The EBV latency III program, characterized by the expression of all EBNAs and LMPs, is commonly present in tumors developing in immunocompromised patients such as post-transplant lymphoproliferative disorders and AIDS-associated lymphomas [101].

The main activator of the EBV lytic program is the viral transcription factor BZLF1 whose expression is under the negative control of the telomerase through the NOTCH2/BATF pathway [102]. On the other end, the inhibition of telomerase expression triggers a complete EBV lytic cycle and the death of EBV-infected cells.

The EBV-driven cell transformation strongly depends on the ability of latent viral proteins to activate multiple signaling pathways, including mitogen-activated protein kinase (MAPK), c-Jun N-terminal kinase (JNK), PI3K/AKT and NF-κB, and to promote unlimited cell proliferation [102,103]. TERT expression has a key role in the maintenance of EBV latency by preventing the viral lytic cycle in both EBV-immortalized lymphobastoid cell lines and EBV-positive Burkitt’s lymphoma (BL) cell lines [104]. On the other end, TERT inhibition induced the expression of BZLF1 and the activation of lytic viral replication leading to the death of EBV-infected cells [104]. Additionally, Giunco et al. observed that short-term inhibition of TERT in EBV-infected cells induced the expression of lytic viral proteins, including BZLF1, and the activation of a complete viral lytic cycle as well as promoting apoptosis via the AKT1/FOXO3/NOXA (AKT-serine/threonine kinase/forkhead box o3/NADPH oxidase activator) and ATM/ATR/TP53 (ATM serine/threonine kinase/ATR serine/threonine kinase/tumor protein P53) pathways [105].

Several studies have demonstrated that EBV LMP1 is able to activate TERT expression at the transcriptional level in both epithelial cells and B lymphocytes by distinct mechanisms [10,106]. Indeed, the transfection of LMP1-expressing vectors in primary human nasopharyngeal epithelial cells caused an increase in telomerase activity related to enhanced MYC expression driven by LMP1 (Figure 2). On the other end, LMP-1 expression in B lymphocytes has been shown to activate TERT via the NF-κB and MAPK/ERK1/2 pathways independently from MYC expression [106,107].

Moreover, LMP1 has been shown to modulate telomerase activity at a post-transcriptional level by inducing the direct binding of TERT to NF-κB p65 and nuclear translocation of TERT in nasopharyngeal carcinoma cells [108].

The EBV latent membrane protein 2A (LMP2A) has been also implicated in the maintenance of the viral latency, but with a negative regulatory effect on TERT expression [109]. Indeed, epithelial cells expressing LMP2A showed a significant reduction in TERT mRNA levels together with decreased telomerase activity. The TERT promoter was inhibited by LMP2A expression both in B cells and epithelial cells, and the tyrosine-based activation motif ITAM (immunoreceptor tyrosine-based activation motif) in LMPA protein was required for such inhibition. Repression of the TERT gene by LMP2A was suggested to hamper B-cell activation and to promote virus latency [109].

Remarkably, TERT inhibition in EBV-positive Burkitt’s lymphomas and lymphoblastoid cell lines caused an enhancement of the apoptotic effect induced by the antiviral therapy-opening opportunities for new therapeutic protocols including TERT inhibitors to treat EBV-related malignancies [105].

### 2.3. Human Herpesvirus 8 and Telomerase Activity in Kaposi Sarcoma

Kaposi’s Sarcoma-Associated Herpesvirus (KSHV) or human gamma herpesvirus 8 (HHV8) has a linear double-stranded DNA genome of approximately 165 kb containing 100 ORFs. Several viral genes share sequence homology to the closely related EBV as well as to cellular genes, while other ORFs are unique with no similarity to other herpesviruses [110]. The HHV8 is recognized as the causative agent of three different types of malignancies such as Kaposi’s sarcoma (KS), multicentric Castleman’s disease (MCD) and a form of AIDS-related primary effusion lymphoma (PEL) [111,112,113]. KS comprises four different clinical forms, including classic KS, prevalent in the Mediterranean region, African-endemic KS that is the most common cancer in the equatorial African countries, iatrogenic KS developing in organ transplant patients and epidemic KS that was the most frequent cancer in AIDS patients before the HAART era [114,115]. Globally, more than 34,000 cases of Kaposi’s sarcoma (KS) were estimated in 2020, all attributable to HHV8 and mainly affecting HIV-positive subjects (71.4%) [116]. MCD and PEL are rare malignancies usually associated with HIV infection.

KS cells are characterized by long telomeres and prominent telomerase activity directly related to the cell proliferation index, indicating that the KS immortalized cells are addicted to TERT activity [117].

Several proteins encoded by conserved or unique genes have been implicated in KS pathogenesis, including K1, K2, vMIPS, K4, K4.1, K5, K9, K12, ORF-6, ORF-71, ORF-72, ORF-73, ORF-74, and K15 [118]. Among these, LANA and v-cyclin, encoded by ORF-72 and ORF-73, respectively, have been shown to play central roles in deregulating various cellular functions. LANA, which is constitutively expressed in HHV8-associated human malignancies, is a functional homolog of EBV EBNA1 protein, acts as an adaptor molecule for an E3 ubiquitin complex via a specific protein motif and causes ubiquitylation and degradation of p53 [119]. LANA also inhibits the Rb tumor suppressor pathway, allowing the infected cells to become resistant to anti-growth signals and cell-cycle arrest [120,121].

Early studies reported that infection of human primary endothelial cells with purified HHV8 particles caused long-term proliferation and cell survival associated with the acquisition of telomerase activity and anchorage-independent growth [122]. LANA has been recognized to act as a transcription factor modulating the regulatory regions of several cellular and viral genes, including the TERT promoter [8]. In particular, LANA has been shown to interact with responsive elements localized in the TERT promoter region containing the five SP1 transcription factor-binding sites. The interaction of LANA with SP1 binding sites has been determined by electrophoretic mobility shift assays using a probe containing GC-boxes in the presence of BJAB nuclear extracts. The lack of interaction between LANA and the GC-boxes in the absence of BJAB nuclear extracts suggested that LANA targets the DNA-protein complex bound to GC-boxes to activate TERT promoter [8]. Furthermore, Verma et al. observed that the transactivation of TERT promoter was due to the physical interaction between LANA N-terminal domains and the target DNA binding site [123].

Proteomic screening experiments allowed the discovery of novel interactions between LANA and cell proteins that could be also involved in TERT [124]. Among sixty-one proteins interacting with adenovirus-expressed Flag-LANA a total of three proteins, namely replication protein A1 (RPA1) and RPA2, xeroderma pigmentosum complementation group A (XPA) and TRF1, were found to associate with LANA and to be involved in the regulation of telomerase activity and telomere maintenance [124]. RPA1 and RPA2 bind single-stranded DNA by forming a complex with XPA protein and playing a critical role in DNA replication, DNA damage response and telomere length-maintenance [125]. In the absence of LANA, the RPA1 and RPA2 proteins localize to telomeric DNA, but are rapidly displaced by the LANA expression suggesting that LANA might inhibit telomere lengthening in cells lacking an alternative lengthening of telomeres (ALT). The analysis of telomere length in HHV8-positive primary effusion lymphoma cell lines BCBL1 and in BC3 revealed shorter telomeres compared to the telomere length in B cells. Therefore, LANA-dependent shortening of telomeres may represent an additional mechanism of cellular transformation [124].

A further effect of HHV8 infection is the induction of ALT-like features, which trigger telomere extension via recombination and break-induced replication, and a concomitant reduction in telomerase levels in non-ALT BJAB, SLK and EA.hy926 cell lines [126]. ALT activity has also been observed in HHV8-associated Kaposi’s sarcoma. Notably, recombination and break-induced replication mechanisms have been shown to be necessary for the survival of HHV8-infected cells, maintenance of viral latency and for replication of HHV8 [126]. The identification of the HHV8 proteins responsible for ALT induction and the mechanisms involved in host-cell alteration could be important for the use of ALT-targeted therapies in KS, including those already available for the treatment of hematological and solid tumors.

### 2.4. Hepatitis B Virus and Telomerase Activity in Hepatocellular Carcinoma

HBV is a small virus belonging to the hepadnavirus family, which contains a 3.2 kb circular double-stranded DNA genome encoding the reverse transcriptase/polymerase (Pol), the capsid protein (core antigen HBcAg), three envelope proteins (L, M, and S) and the transactivating protein x (HBx) [127].

HBV infects more than 2 billion people worldwide, of whom 240 million develop a chronic infection and are at elevated risk of cirrhosis and hepatocellular carcinoma (HCC) [128,129]. The incidence of HCC among people with chronic HBV infection ranges from 400 to 800 per 100,000 males/year and from 120 to 180 per 100,000 females/year [130].

Telomerase activity and telomeres length have crucial roles in hepatocarcinogenesis [131]. TERT expression in normal hepatocytes gradually decreases during embryo development and it is fully inhibited in the adult hepatocytes which show non-shortened telomeres and several markers of senescence [131]. During cirrhosis, telomeres become very short as a consequence of liver damage and unceasing cell renewal, thus causing the activation of DNA damage response, cell senescence and activation of telomerase [132].

TERT expression and telomeres’ elongation are detected in above 90% of HCC and reported to be associated with tumor aggressiveness and worse patient prognosis [133]. Major mechanisms of telomerase activation in HCC involve the occurrence of somatic mutations in the TERT core promoter, TERT gene amplification and HBV genome integration nearby the TERT loci [30,134,135,136,137,138]. The TERT promoter region has been identified as the most frequent site of HBV insertion in HCC (38.5%), but it is rarely observed in non-tumor liver tissues (3%) [139].

Two proteins encoded by HBV, namely HBx and surface (S) gene (preS2), have been shown to increase TERT expression and telomerase activity in hepatoma cells and in HCC cell lines, respectively [140,141]. In particular, wild type and truncated forms of HBx proteins have shown to increase telomerase activity and TERT expression in a dose-dependent manner in different HBx-transfected cells. The HBx responsive region has been localized in TERT core promoter (nt −132 to nt +5) and the HBx transactivation mechanism relies on its ability to stabilize SP1 to binding sites, as demonstrated by electrophoretic mobility shift assays [140]. HBx is highly expressed in HBV-related HCC biopsies and it is associated with increased transcription of TERT gene [140].

The HBV preS2 has also been shown to activate TERT expression, telomerase function, and to enhance the malignancy of HepG2 cells in vivo and in vitro [141]. Suppression of preS2 expression caused a decrease in TERT mRNA levels and telomerase activity as well as inhibition of cell proliferation and tumorigenicity of HepG2 cells. The preS2 was demonstrated to activate the transcription of TERT gene via a 20 bp long preS2-responsible region (PRR) located between nt −349 and nt −329 bp upstream of TERT transcription start site [141].

Moreover, a component of the telomerase complex, namely NHP2, has been found overexpressed along with TERT enzyme in HBV-related HCC as well as in HBx-transduced hepatoma cell lines. The silencing of the NHP2 gene caused a decrease in of TERT expression, destabilization of the telomerase complex and inhibition of the proliferation of hepatoma cells overexpressing HBx [142]. The knockdown of NHP2 also suppressed HBx-transduced hepatoma cell growth in a xenograft animal model, suggesting that therapeutic approaches targeting NHP2 may provide a novel strategy for treating HBV-related HCC [142].

Effective treatment options for HCC are very limited and the well-known key role of telomerase in liver cancer development makes it a promising therapeutic target. New strategies to target telomerase in liver cancer are being studied in vivo and in vitro and include small molecules inhibiting TERT enzymes, antisense oligonucleotides and molecules that stabilize the DNA G-quadruplex or block telomerase access to telomeres [143].

### 2.5. Hepatitis C Virus and Telomerase Activity in Hepatocellular Carcinoma

HCV, classified in the flaviviridae family, has a single-stranded RNA genome encoding a 3000-amino acid polyprotein which is cleaved by viral and cellular proteases into four structural proteins, named capsid protein C, envelope glycoproteins E1 and E2, and protein P7, and six nonstructural proteins, named NS2, NS3, NS4A, NS4B, NS5A and NS5B [144].

HCV represents a major health problem in the world with over 185 million infected people and about 80–85% evolving into chronic infections with the risk of causing cirrhosis, HCC or lymphoproliferative disorders [145]. The antibody levels against specific peptides of the HCV proteome, including the N-terminal HCV core peptides, have been found to be significantly high in HCC patients and proposed as useful biomarkers of disease progression among HCV-positive patients [146]. HCC is the second leading cause of cancer-related death worldwide and HCV infection has been shown to contribute to 80% of HCC cases [147]. In addition, HCV may infect cells other than hepatocytes and it has been implicated in extrahepatic neoplasia such as non-Hodgkin lymphoma, cholangiocarcinoma, pancreatic cancer and oral carcinomas

Similarly to HBV-related HCC, HCV-related cancers also show high TERT expression and telomerase activity which is associated with tumor progression and aggressiveness [143]. HCV core protein is considered a positive regulator of telomerase function. Indeed, the expression of HCV core protein in human hepatoma cells has been reported to increase TERT gene transcription, telomerase activity and localization of TERT in the nucleus [148]. Although the molecular mechanisms involved in the telomerase regulatory properties of core protein are not known, this pathway may also contribute to the hepatocarcinogenesis mediated by HCV.

An additional effect of HCV core protein on telomerase activation has been shown to work via downregulation of microRNA-138 (miR-138) in HCV-related HCC [149]. TERT mRNA is a direct target of miR-138 in HCC cells, and then the suppression of miR-138 by the mature HCV core protein (173 amino acids long) causes an increase in telomerase expression, inhibition of cell replicative senescence and promotion of hepatocarcinogenesis [149].

An additional mechanism of HCV-related telomerase deregulation is based on the ability of NS3-4A protease-helicase to bind the C-terminal region of TERT. The NS3-4A/TERT complex triggers increased telomerase activity in NS3-4A-transfected cells [150]. Such an effect was hindered by NS3-4A helicase and protease inhibitors proving its key role in the regulation of the telomerase and cell transformation.

The need for clinical trials evaluating the efficacy of anti-telomerase compounds in the treatment of HCC has been recently prompted by preclinical studies showing that liver cancer cell lines have oncogenic addiction to TERT expression and that malignant phenotype may be reverted by anti-TERT antisense oligonucleotides treatment. Current therapeutic approaches now should aim to target telomerase and other oncogenic pathways coexisting with TERT activation in specific subgroups of HCC. Telomerase inactivation will remain, for the coming years, a mainstay for HCC treatment either in preclinical studies or in clinical trials.

Somatic mutations in the core promoter of TERT gene have recently emerged as the most prevalent mechanism of telomerase activation in HCV-related HCC [30]. Telomeres length and TERT mRNA expression in HCC and peri-tumor tissues are associated with distinct clinical characteristics [151]. Importantly, liver cancer cell lines have been shown to be addicted to TERT and that malignant phenotype may be reverted by anti-TERT oligonucleotides [151]. These results suggest that telomerase inhibition may be an effective therapeutic strategy for HCV-positive HCC.

### 2.6. Human T-Cell Leukemia Virus Type 1 and Telomerase Activity in T-Cell Lymphoproliferative Disorders

HTLV1 was the first oncogenic human retrovirus to be discovered. HTLV1 contains a 8.5 kb linear, single-stranded RNA-positive genome, encoding structural proteins and enzymes as well as regulatory and accessory factors [152,153]. HTLV1 causes lifetime infection in about 10 million people worldwide, which, following an incubation period lasting 15–30 years, may impair the immune system causing lymphoproliferative diseases such as adult T-cell leukemia and HTLV1-associated myelopathy/tropical spastic paraparesis [154].

Early studies reported strong telomerase activity, despite the maintenance of short telomeres, in HTLV1-positive adult T-cell leukemia cells compared to asymptomatic carriers or normal donors [155]. Moreover, the telomerase activity was reported to gradually increase with progression from the asymptomatic stages to acute and chronic disease [156,157].

Infection of human primary T cells with HTLV1 has been shown to cause transcriptional activation of TERT gene [158]. Moreover, Tax protein, transiently transduced into primary lymphocytes, was able to induce telomerase expression through the nuclear NF-κB signaling pathway. The NF-κB-dependent activation of the TERT promoter is based on the increased binding of c-MYC and SP1 to the cognate binding sites in HTLV1 and Tax-expressing cells [158].

The shelterin complex is known to regulate telomeres’ length by controlling the access of telomerase to the ends of chromosomes [159]. The TRF1, TRF2 and TIN2 factors of the shelterin complex are overexpressed in HTLV1-infected adult T-cell leukemia cells and are probably responsible for the maintenance of short telomeres and apoptotic inhibition [155,160].

An alternative mechanism of regulation of TERT promoter activity is based on the balance between the inhibitory function of Tax protein and the activating effect of the HBZ proteins (HTLV1 basic leucine zipper) [161]. In this model, the expression of Tax early after infection is supposed to inhibit TERT expression favoring telomeres attrition, genomic instability and neoplastic progression, while a later stage Tax may be repressed and the HBZ expression reactivates telomerase function allowing progression toward fully cell transformation. Furthermore, HBZ has been shown to cooperate with JunD proto-oncogene, AP-1 transcription Factor subunit (JUND), to enhance TERT transcription in adult T-cell leukemia cells [162]. The tumor suppressor menin (MEN1) is able to interact with JUND and to negatively regulate telomerase expression. However, in HBZ-expressing cells the JUND/MEN1 complex binds to HBZ and activates the histone acetyltransferase p300 leading to a reduced JUND/MEN1 suppressor activity and increased TERT transcription [162]. Furthermore, the HBZ has been found to trigger proteasome-mediated degradation of TAL BHLH transcription factor 1, erythroid differentiation factor (TAL1) factor, which is a major TERT gene repressor in T lymphoblasts, thus inducing TERT expression and telomerase activity at later stages of infection [161].

In conclusion, HTLV1 infection significantly alters both telomerase activity and TERT expression by multiple mechanisms depending on the infectious status, the level of Tax and the activation stage of T cells. On the other end, the HTLV1-related telomere modulation may be involved in T-cell clonal expansion, immunosuppression and tumor development [163].

## 3. Telomerase and Immune Response in Virus-Related Cancers

Virus-related tumors develop in a small percentage of infected subjects which fail to produce an effective immune response against infected cells. Telomerase expression has been recently implicated in the promotion of the immune suppressive tumor microenvironment.

Effective immune response to viral antigens involves the temporarily upregulation of telomerase activity and the clonal expansion of virus-specific T cells. The reduced proliferation of these cells may critically delay the T-cell immune function [164]. However, in chronic viral infection, the repeated stimulation of CD8+ T cells causes the loss of telomerase reactivation, while CD4+ T cells maintain the telomerase activity after many antigenic stimulations [165]. Exhausted viral-specific T cells have been described for both HBV and HCV and have been found in the liver, spleen and blood.

Moreover, the study of the gene expression profiles of more than 9000 tumors, including virus-related cancers, showed that TERT mRNA levels were associated with immune suppressive signatures via TERT-dependent activation of endogenous retroviruses, which by forming double-stranded RNAs induced the RIG-1/MDA5-MAVS signaling pathway, interferon signaling, chemokines’ expression and recruitment of suppressive T cells in the tumor [166].

Therefore, it is possible to hypothesize that telomerase may potentiate the oncogenic effect of oncoviruses by a positive feedback loop in which viral factors induce TERT activation and TERT in turn causes the expression of endogenous retrovirus mRNAs that inhibit the immune surveillance on cancer-infected cells.

## 4. Conclusions

The human oncoviruses share similar molecular mechanisms to promote cell transformation [167]. The activation of telomerase represents a common strategy operated by almost all oncoviruses to evade replicative senescence and to increase the proliferative capacity of infected cells (Table 1). In addition, telomerase has a crucial role in the regulation of the replicative program of some viruses and in the maintenance of cell immortality. Indeed, TERT expression is a main regulator of EBV latency and cell transformation, whereas TERT inhibition induces the lytic cycle of EBV and cell death. The regulation of telomerase activity by virus-encoded factors is complex and occurs at multiple levels, such as transcription, alternative splicing, post-transcription modifications and subcellular localization. Numerous studies have shown that TERT, beyond its telomere-lengthening function, concurs with cancer development via multiple activities, also including the regulation of T-cell function particularly in virus-infected cells [168,169].

The Merkel cell polyomavirus is the only oncovirus which has not been implicated in TERT reactivation. However, the early proteins encoded by a recently discovered polyomavirus, namely Lyon IARC polyomavirus (LIPyV), have been demonstrated to activate TERT gene expression, via recruitment of the SP1 transcription factor to the TERT promoter [170]. Therefore, TERT activation may be considered a common path in virus-related tumorigenesis.

We and others have shown that TERT promoter mutations, causing hyperactivation of telomerase, are highly frequent in HPV-related cancers and HCV-related hepatocellular carcinoma [28,31,32,135,171]. In particular, all TERT promoter mutations are G>A transitions, which generate novel E-twenty-six binding sites, affect chromatin looping, interfere with the interaction of telomeres with the TERT loci as well as destabilizing the DNA secondary structures and G-quadruplexes formed within the TERT promoter [171]. It is not yet known whether these mutations are also capable of potentiating the transactivation of the TERT promoter by oncoviral proteins. Of note, a recent study showed comparable levels of HPV16 E6 and a significant increase in TERT mRNA in cervical squamous cell carcinoma specimens mutated in the TERT promoter versus those not mutated, suggesting a synergistic effect of mutations and viral oncoproteins in telomerase expression [32].

Further investigation on the telomerase and oncoviruses interplay is conceivable to yield new advances and new therapeutic strategies to treat virus-related human cancers.

## Figures and Tables

**Figure 1 cancers-14-05257-f001:**
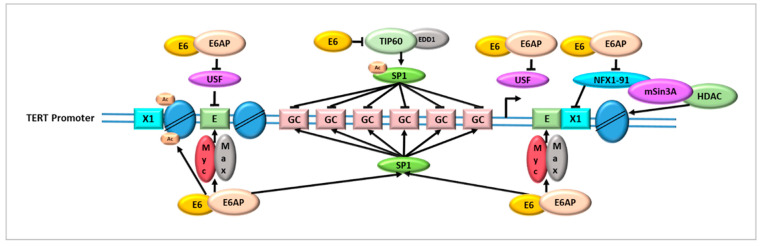
The main strategies adopted by HPV E6 to regulate telomerase activity at transcriptional level relies on the ability of HPV E6/E6AP complex to stabilize MYC/MAX and SP1 transcription activators, to dislocate USF and NFX1-91 transcription repressors, to induce the acetylation of histone H3 and to inhibit TIP60-mediated acetylation of SP1.

**Figure 2 cancers-14-05257-f002:**
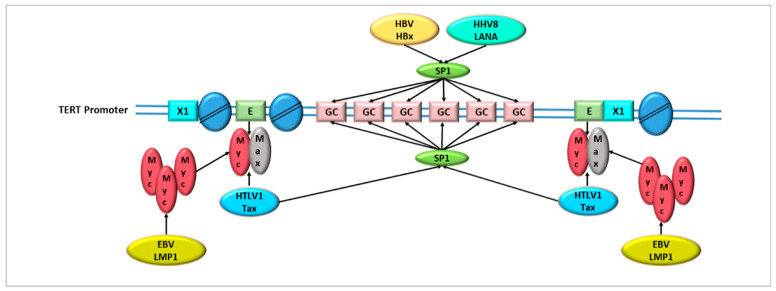
Common strategies adopted by oncoviruses to regulate telomerase activity at transcriptional level: HHV8 LANA and HBV HBx are able to stabilize the binding of SP1 to GC-binding sites, HTLV1 Tax facilitates the binding of MYC/MAX to E-boxes and EBV LMP1 recruits MYC on the cognate binding sites on TERT promoter.

**Table 1 cancers-14-05257-t001:** Mechanisms of interaction between oncoviruses and telomerase in tumors associated with infectious agents.

Oncovirus	Cancer	Viral Protein	Mechanisms of Interaction with Telomerase Function	Reference
HPV	Cervix, anus, vagina, vulva, penis, head and neck	E6	Induces high levels of telomerase	[9,64]
			The complex E6/E6AP stabilizes MYC/MAX heterodimers on the E-boxes	[65,70,71,72,73]
			The complex E6AP binds and degrades the TERT promoter negative regulator NFX1-91 causing TERT overexpression	[74,75,76,77]
			Cooperates with NFX1-123 and PABPCs in the post-transcriptional stabilization of TERT mRNA	[79,80]
			The complex E6/E6AP causes the acetylation of H3 at TERT promoter and increased transcription	[77]
			E6 destabilizes the tumor suppressor TIP60 causing the accrual of SP1 on GC boxes	[82]
			Directly interacts with TERT protein and associates with telomerase complexes and telomeres	[61]
		E7	Potentiates the effect of E6 on telomerase upregulation	[65]
EBV	Burkitt’s lymphoma, nasopharyngeal carcinoma	LMP1	Activates TERT via the NF-κB and MAPK/ERK1/2 pathways independently from MYC expression	[106,107]
			Induces the binding of TERT to NF-κB p65 and nuclear translocation of TERT in nasopharyngeal carcinoma cells	[108]
		LMP2A	Has a negative regulatory effect on TERT RNA expression and promotion of virus latency	[109]
HHV8	Kaposi’s sarcoma, multicentric Castleman’s disease, primary effusion lymphoma	LANA	Interacts with the protein complexes binding to SP1 responsive elements localized in the TERT promoter	[8]
			Associates with RPA1, RPA2, XPA and TRF1 causing their displacement from telomeric DNA	[124]
		?	Induction of ALT-like features in non-ALT cells infected with HHV8, but the viral factors involved are not known	[126]
HBV	Hepatocellular carcinoma	HBx	Stabilizes the binding of SP1 protein to the GC boxes	[140]
		preS2	Binds to a 20 bp long preS2-responsible region (PRR) located between nt -349 and nt -329 bp upstream of TERT transcription start site	[141]
HCV	Hepatocellular carcinoma and lymphoproliferative disorders	core	Induces TERT gene transcription, telomerase activity and localization of TERT in the nucleus in hepatoma cells, however the molecular mechanisms are not known	[148]
			Downregulates microRNA-138, which targets TERT mRNA, causing telomerase expression	[149]
		NS3-4A	Binds the C-terminal region of TERT and the NS3-4A/TERT complex promotes telomerase activity in transfected cells	[150]
HTLV1	T-cell leukemia and HTLV1-associated myelopathy/tropical spastic paraparesis	Tax	Induces telomerase expression through the nuclear NF-κB signaling pathway	[158]
			Stabilizes the binding of MYC and SP1 to the cognate binding sites	[158]
		HBZ	Cooperates with JunD proto-oncogene, AP-1 transcription Factor subunit (JUND) to enhance TERT transcription in adult T-cell leukemia cells	[162]
			Activates the histone acetyltransferase p300 leading to a reduced JUND/MEN1 suppressor activity and increased TERT transcription	[162]

## Data Availability

The original contributions presented in the study are included in the article. Further inquiries can be directed to the corresponding author.

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
