# Peer review of "The Molecular Interplay between Human Oncoviruses and Telomerase in Cancer Development"

_cancers, 2022, doi:10.3390/cancers14215257_

Round 1
Reviewer 1 Report
In the Manuscript by Tornsello et al., the authors made a review of literature data on the molecular interplay between human oncoviruses and telomerase in cancer development. They illustrated Telomerase reverse transcriptase (TERT) expression and activities, then they browsed through the different human oncoviruses, and they described the ability of each of them to interact/interfere with TERT.
The review is well conceived, structured, and written ad it is easily readable even for a non-specialized reader; the sections are complete and give a good landscape of the literature data for the relevant topic.
I’ve just to report two typos in the MS:
1- Line 411: “HTLV1 associated meylopathy/tropical spastic parapresis” has to be changed into “HTLV 1 associated myelopathy/tropical spastic paraparesis”
2- Check and uniform throughout the manuscript the acronym “NF-κB” because sometimes it is well written whereas some in other parts is written “NF-kB” (es. Line 420) or “NFkB” (es. Line 421 in “NFkB-dependent…”)
Collectively, I consider the MS by Tornesello et al., suitable for publication in Cancer in this form, after the typos revision.
Author Response
We thank the reviewer for the positive comments on the manuscript. The suggested corrections have been made in the text.

Reviewer 2 Report
This review, "The molecular interplay between human oncoviruses and telomerase in cancer development,” shows evidence of the main functions of telomerase in the progression of various cancers and the role of viral proteins of some oncoviruses that affect at different levels the expression of telomerase and thereby induce processes associated with the progression of multiple cancers. The review seems very important to me. However, I have some suggestions:
Two central themes in which the writing is developed are noted: the role of telomerase and the molecular mechanisms associated with its deregulation in cancer progression and the evidence showing the effect of various viral oncoproteins in the deregulation of molecular mechanisms that impact TERT overexpression and telomere elongation, however, the subthemes should be handled as such.
The sections describing oncoviruses should have a similar format: general characteristics of oncoviruses, epidemiology of viral infections associated with cancer(s), telomerase expression levels in those types of cancer, the findings obtained on the molecular mechanisms linked to telomerase overexpression at the level of transcriptional regulation, of the involvement of oncoproteins in TERT overexpression, etc.
In conclusion, they do not conclude that they place information linking TERT overexpression and evasion of the immune response; I would suggest that if there is enough information, it should be considered an additional topic.
The figure, I think, should, rather than showing the independent mechanisms, could make an effort to integrate the common mechanisms in terms of regulation of TERT expression, activation, and functions in the development of oncovirus-associated cancers described in the review.
Author Response
We thank the reviewer for positive comments as well as remarks. All suggestions have been taken into account in the revised version of the manuscript. All changes have been highlighted in red colour
